

# Flooding-related increases in $CO_2$ and $N_2O$ emissions from a temperate coastal grassland ecosystem

Amanuel W. Gebremichael[1,2], Bruce Osborne[1,2], Patrick Orr[1]

[1]UCD School of Earth Sciences, University College Dublin, Belfield, Dublin 4, Ireland
[2]UCD School of Biology and Environmental Sciences, University College Dublin, Belfield, Dublin 4, Ireland

*Correspondence to*: Amanuel W. Gebremichael (amanuel.gebremichael@ucdconnect.ie)

**Abstract**

Given their increasing trend in Europe, an understanding of the role that flooding events play in carbon and nitrogen cycling and greenhouse gas (GHG) emissions will be important for improved assessments of local and
regional GHG budgets. This study presents the results of an analysis of the $CO_2$ and $N_2O$ fluxes from a coastal grassland ecosystem affected by episodic flooding that was of either a relatively short or long duration (SFS and LFS sites, respectively). Compared to the SFS, the annual $CO_2$ and $N_2O$ emissions were 1.4 and 1.3 times higher at the LFS, respectively. Mean $CO_2$ emissions during the period of standing water were $144 \pm 18.18$ and $111 \pm 9.51$ mg $CO_2$-C m$^{-2}$ h$^{-1}$, respectively, for the LFS and SFS sites. During the growing season, when there was no
standing water, the $CO_2$ emissions were significantly larger from the LFS ($244 \pm 24.88$ mg $CO_2$-C m$^{-2}$ h$^{-1}$) than the SFS ($183 \pm 14.90$ mg $CO_2$-C m$^{-2}$ h$^{-1}$). Fluxes of $N_2O$ ranged from -0.37 to 0.65 mg $N_2O$-N m$^{-2}$ h$^{-1}$ at the LFS and from -0.50 to 0.55 mg $N_2O$-N m$^{-2}$ h$^{-1}$ at the SFS, with the larger emissions associated with the presence of standing water at the LFS and during the growing season at the SFS. Overall, soil temperature and moisture content were identified as the main drivers of the seasonal changes in $CO_2$ fluxes, but neither adequately
explained the variations in $N_2O$ fluxes. Analysis of total Carbon (C), Nitrogen (N), microbial biomass and $Q_{10}$ values, indicated that the higher $CO_2$ emissions from the LFS were linked to the flooding-associated influx of nutrients and alterations in soil microbial populations. These results demonstrate that annual $CO_2$ and $N_2O$ emissions can be higher in longer-term flooded sites that receive significant amounts of nutrients and where diffusional limitations due to the presence of standing water is limited to periods of the year when the
temperatures are lowest.




## 1 Introduction

The frequency of flooding events has increased in Europe in the last three decades, and is likely to increase further in a warmer climate, as a consequence of climate change (Beniston et al., 2007; Christensen and Christensen, 2007). Although flooding could have significant implications for greenhouse gas (GHG) emissions due to the effects of a water barrier on gaseous diffusion, as well as through alterations in soil biological and physio-chemical processes (Hansen et al., 2013; Peralta et al., 2013), this has rarely been assessed. It is therefore unclear how flooding influences annual ecosystem GHG budgets, particularly in flood-prone ecosystems that experience variable periods of inundation. The impact of flooding will depend on the timing and duration of the period of inundation, as well as on site-related characteristics related to topography and hydrology. The extent of fresh water flooding will also depend on the frequency and magnitude of rainfall events.

The natural topographic and hydrological conditions of many coastal plain or floodplain wetlands increases the possibility of these ecosystems being rapidly inundated with freshwater following extreme precipitation events or extended periods of rainfall. Episodic flooding is a common feature of these ecosystems, with potential sources of water from inland or inshore incursions, although inundation from freshwater sources is probably more common (Doornkamp, 1998). Many assessments of GHG emissions, and an understanding of the factors that control them, are based on information from ecosystems less prone to flooding episodes. This data may not be directly applicable to coastal areas or floodplains where both the volumes involved and the frequency and duration of water incursions can be high. It is well known that soil water availability can affect the emissions of $CO_2$ and $N_2O$ by influencing the rates of C and N mineralization (Alongi et al., 1999; Noe et al., 2013) but less clear how variations in standing water levels influence these processes, although this is likely to depend on the extent and the duration of inundation (Lewis et al., 2014). The literature available provides contrasting evidence as to whether soil inundation enhances or impedes organic matter mineralization. Wilson et al., (2011), and Kim et al., (2015) showed the rate of organic matter mineralization can be enhanced after a short hydroperiod (the period the soil area was waterlogged), whereas longer inundation periods suppressed decomposition by limiting oxygen supply (Lewis et al., 2014). Flooding generally promotes anaerobic conditions by excluding air from the pore spaces in the soil that would reduce the mineralisation of organic matter (Altor and Mitsch, 2008). Despite lower mineralisation rates, under these conditions, $CO_2$ emissions still remain possible from anoxic soils (Glatzel et al., 2004). A number of factors, such as the quality and quantity of substrates available for microbial processes, temperature, soil microbial activity, and oxidation-reduction potential, could impact on the magnitude of $CO_2$ or $N_2O$ exchange both in flooded and non-flooded ecosystems. High soil organic matter concentrations, in combination with warmer conditions in flooded soil, can also enhance $CO_2$ production (Oelbermann and Schiff, 2008; Kim et al., 2015). When associated with sufficient oxygen, $CO_2$ emissions in wetlands increase as a result of accelerated organic matter decomposition, whilst $CH_4$ production decreases because of aerobic methane oxidation (Smith et al., 2003). Whilst $CH_4$ is often considered to be a much more significant GHG in permanently wet or flooded ecosystems, this may not be the case where there are only temporary hydroperiods (Audet et al., 2013; Batson et al., 2015), where $CO_2$ fluxes are still likely to dominate the annual budget. In addition, the effect of temporary or contrasting hydroperiods on $N_2O$ fluxes has received little, if any, consideration. For this reason, we focussed on these two GHGs, which are regarded as the more important ones in many terrestrial ecosystems.





Denitrification and nitrification are often considered as the main mechanisms by which $N_2O$ is produced in soil; although there is evidence that denitrification may be the dominant process for the release of $N_2O$ both under aerobic and anaerobic soil conditions (Bateman and Baggs, 2005). Whilst denitrification and increased $N_2O$ production is enhanced by high soil water contents (Machefert and Dise, 2004; Maag and Vinther, 1996), the

frequency and duration of flooding may also be contributory factors controlling the evolution of $N_2O$. In riparian systems, a shorter flooding duration produced the highest $N_2O$ emissions, which diminished the longer the period of inundation (Jacinthe et al., 2012). In addition to water availability, temperature is also an important variable in determining the production and consumption of $N_2O$ and $CO_2$, by affecting the metabolic activity of microorganisms and plants (Davidson and Janssens, 2006; Butterbach-Bahl et al., 2013; Kirwan et al., 2014;

Kim et al., 2015).

Flooding stimulates changes in the structure of soil microbial communities (Bossio and Scow, 1998; Unger et al., 2009; Wilson et al., 2011), and this, in turn, affects the rate of decomposition of organic material (Van Der Heijden et al., 2008). The extent of any change in microbial biomass and/or microbial populations could also vary with flooding duration (Rinklebe and Langer, 2006). For example, Wilson et al., (2011) showed significant

changes in microbial structure and increases in soil enzymatic activity after the short-term (24 days) inundation of floodplain soil. Nevertheless, an increase in microbial activity and GHG emissions even after short-term flooding is not commonly accepted (Unger et al., 2009; Jacinthe, 2015), and the timing as well as the frequency of flooding events may be important in determining both the microbial community change and the associated GHG emissions. Organic substrate availability (through the breakdown of plant litter or organic compounds

transported by flowing water) is considered a key driving factor regulating the activities of soil microorganisms and the mineralization and immobilization of carbon and nitrogen (Badiou et al., 2011; Li et al., 2015). The availability of organic substrates influences the activity of carbon-cycling extracellular enzymes and, as a result, can impact on the rate of $CO_2$ emissions and potentially other GHGs (Li et al., 2015).

Owing to the episodic nature of flooding events, capturing their impact on GHG emissions is quite challenging

and this could perhaps be the reason for the lack of data from field studies. Given the projected increase in flooding events, an evaluation of their impact on GHG emissions is clearly required. Identification of sites with flooding potential before flooding events, and the establishment of the appropriate measurement protocols are crucial steps required to capture the dynamics of GHG fluxes in response to real flooding events. This would enable the capture of the full pattern of GHG fluxes prior to, during and after a flooding event. In this paper, we

assess the impact of freshwater inundation, with different hydroperiods, on $CO_2$ and $N_2O$ fluxes before, during and after real flooding events in a coastal grassland ecosystem over ~2 years. The main objective was to assess the effects of short- and long-term flooding on the dynamics of $CO_2$ and $N_2O$ fluxes and how this impacts on the annual budgets for each gas. We also evaluate the significance of a number of soil physical, chemical and biological parameters, which may influence the fluxes of $CO_2$ and $N_2O$.





## 2 Materials and methods

### 2.1 Site description

The study site (53°05'87" N, 6°04'07" W) was situated on a low lying area (~ 0 m a.s.l)of coastal grassland that forms part of the East Coast Nature Reserve (ECNR), a portion of the larger coastal wetland complex called the Murrough wetlands on the east coast of Ireland, near Newcastle, Co. Wicklow (Fig. 1). The site is owned and managed by BirdWatch Ireland, predominantly as a habitat for a wide variety of birds. The grassland area lies between a long drainage ditch that runs in a north-south direction and a shingle beach bordering the Irish Sea. In the past 12 years, restoration methods, including water management, low intensity grazing and crop planting, have been undertaken to maintain the site. The standing water level varies spatially across the site and fluctuates seasonally in response to rainfall and the water-retaining capacity of the drainage ditch. The site experiences near to complete saturation with localised flooding in autumn and winter, but drains and dries out in spring and summer. Despite being located in a coastal area, the ditch water is generally characterized as being fresh water with most sourced from the landward side.

Differences in elevation (~20-25 cm vertical height) over the site result in variations in hydrological connectivity between different portions of the grassland and the ditch water system. Initial monitoring of the general area indicated that this resulted in spatial hydroperiodic differences. The site was therefore divided into two areas based on these hydroperiod characteristics. The site on the more elevated ground is characterized by less frequent and shorter term flooding (SFS) whilst the site located at lower elevations becomes inundated seasonally for an extended period during the winter (longer term flooding: LFS). The plant community of the SFS is mainly dominated by Creeping Bent (*Agrostis stolonifera*) (48%), Velvet grass (*Holcus lanatus*) (27%), Hair grass (*Eleocharis acicularis*) (11%), Meadow grass (*Poa*) (10%) and Couch grass (*Elymus repens*) (7%). The LFS is dominated by grasses and rushes, including Common Couch Grass (*Elytrigia repens*) (80%), Sharp-flowered rush (*Juncus acutiflorus*) (12%) and Curled Dock (*Rumex crispus*) (4%). The mean annual rainfall of the study area is 756 mm (based on data from the Dublin Airport station, 50 km north of our study site, data source: Met Éireann).

### 2.2 Greenhouse gas measurements

The fluxes of the greenhouse gases, $CO_2$ and $N_2O$, were measured using closed chambers. The chambers were made from 16 cm diameter acrylic tube cut to a length of 23 cm with a flat cap of similar material fitted over the top, with an open base. In order to increase the opacity and reflectivity of the Chambers, they were painted dark grey inside and outside and, additionally, their outer surfaces were wrapped in silver duct tape. The chambers were placed on top of collars (~16 cm diameter), with rubber lips around the top, which were inserted into the soil to 5 cm depth. The purpose of the rubber lip was to create a secure seal between the chamber and the collar during the measurements. After establishment of the LFS and SFS sites in September 2013, 6 collars were installed at each site. Individual collars were approximately 6 m apart and the array staggered along two parallel transects. There was ~ 12 m buffer zone between the two sites. During the period when the sites were inundated, the same chamber was used for sampling, but was inserted into a Styrofoam support enabling it to be balanced on top of the collar by floating on the water surface. The collars were also extended to a height of 20 cm by





fitting another similar 15 cm long tube just before the onset of the flooding period. To prevent the chamber from being displaced by any wind during sampling, four thin rods were fixed into the ground around each sampling point to maintain the chamber in position. Sampling of gases was generally undertaken two to four times each month, but less frequently in the winter months.

Measurements of the $CO_2$ and $N_2O$ concentration inside the chamber were made using a Photoacoustic gas analyser (PAS) (Innova 1412, Denmark), connected to the chamber using Teflon tubing. The tubes were 6 m long with a 4 mm inner diameter and the inlet and outlet of the PAS connected to two ports on the top of the chamber. For sampling, the chamber was placed over the collar for between 5 and 6 minutes during which time the gas concentration was analysed 5 to 7 times to complete one sample. Fluxes of $CO_2$ and $N_2O$ (mg m$^{-2}$hr$^{-1}$)

were calculated using:

$$F = (\Delta C/\Delta t)\ (V/A)$$

Where $\Delta C/\Delta t$ : the rate of change in gas concentration inside the chamber during the chamber placement period, which was calculated by fitting a best fit linear regression line to this data versus time; V : chamber volume (4.069 x 10$^{-3}$ m$^3$); and  A : area bounded by the chamber (0.016 m$^2$). Fluxes of $CO_2$ and $N_2O$ were computed if

linear regressions produced r$^2$ > 0.90 (*P<0.05*) for $CO_2$ and r$^2$ > 0.70 (*P<0.05*) for $N_2O$.

Annual $CO_2$ and $N_2O$ emissions for Feb. 2014-Feb. 2015 and May, 2014-April, 2015 were computed by linear interpolation of fluxes for each sampling date. The area under the curve was calculated using the trapezoid rule by integrating the area for 12 month periods. To estimate and compare the contribution of $CO_2$ and $N_2O$ fluxes to the Global warming potential (GWP), $N_2O$ was converted to $CO_2$-equivalents by multiplying it by 298

(Solomon et al., 2007).

Values of $Q_{10}$ for $CO_2$ emissions were computed for the SFS and LFS using the equation $Q_{10}$ = exp (slope*10), where the slope was derived from the regression coefficient of the exponential equation fitted to the $CO_2$ flux and temperature data.

### 2.3 Environmental measurements

Along with the flux measurements, other environmental variables that could potentially influence the GHG fluxes were also measured.  A weather station, located about ~ 100m from the locality where measurements were made, comprised sensors for air temperature (RHT3nl-CA), humidity (RHT3nl-CA), solar radiation (PYRPA-03) and rainfall (RG2+WS-CA). Average air temperature and cumulative rainfall were recorded at 2 m height every 5, and 60 minutes, respectively. Soil moisture content and temperature were measured adjacent to

the collars/chambers using a hand-held Theta probe (Delta-T Devices Ltd., Cambridge, UK) each time gas sampling was performed. The depth of standing water (WD) was measured adjacent to the gas sampling points using a graduated wooden ruler. Redox potential was measured from each collar using a portable Hanna redox meter (HI9125, Hanna Instruments) with a 10 cm redox electrode. Redox potential was measured from the soil surface except when the LFS was flooded above a height of 10 cm, in which case the measurements were



acquired from the surface of water. Complete insertion of the electrode to the top soil was avoided to prevent the uncertain impact of the intrusion of water into the electrode through the rim at the top during sampling.

### 2.4 Soil sampling for physical and chemical analysis

Sampling of soil from the two sites (each n = 6) for analysis of its physicochemical properties was carried out in July 2015 and the samples were then air dried and sieved (2 mm) before analysis. Soil texture was measured using the pipette method (Gee and Bauder, 1986) by first removing the organic content using hydrogen peroxide and then dispersing the samples with sodium hexametaphosphate. Particles were classified as sand (0.063-2 mm), silt (0.002-0.063 mm) and clay (<0.002 mm). Bulk density at the topsoil was determined by drying each soil sample in a soil corer (each 5 cm diameter x 7 cm height) at 105 $^0$C for 48 hrs. The density was calculated by dividing the dried soil mass by the core volume. Soil pH was determined on 10 g soil dispersed in 20 ml of deionized water; after 10 min equilibration, a reading was taken using a pH probe/meter (Thermo Fisher Scientific Inc., Waltham, Michigan, USA) while stirring the suspension. Total Carbon (TC) and Nitrogen (TC) were determined through combustion of finely ground (0.105 mm sieve size) soil using a LECO TruSpec Carbon-Nitrogen analyser (TruSpec®, LECO Corporation, Michigan, USA). Analysis of TC, TN and pH were performed for samples taken every 5 cm from a 25 cm profile (5 samples per sampling point).

Sampling for analysis of $NO_3$-N and $NH_4$-N (n = 4-6 from each of the LFS and SFS site) was carried out on six occasions from 5 cm depths and an extraction performed by mixing 10 g fresh soil with 2 M KCl. After shaking for 1 hour, the KCl extracts were filtered and stored in the freezer until analysis. $NO_3$-N and $NH_4$-N were determined from the extracts using an ion analyser (Lachat, QuikChem®, 5600 Lindburgh Drive, Loveland, Colorado, USA)

### 2.5 Microbial biomass

Microbial biomass C (MBC) and N (MBN) was determined using the chloroform fumigation extraction method (Vance et al., 1987). 10 g samples of fresh soil (n=4-6) from each site were fumigated in a desiccator with 20 ml ethanol-free chloroform for 72 hours and then extracted with 0.5 M $K_2SO4$. Identical numbers of subsamples were extracted with the same solution but without fumigation the day after sampling. Supernatants from both fumigated and non-fumigated samples were filtered through Whatman No. 1 filter paper and stored in the freezer until analysis. Organic carbon and total nitrogen in the filtrate were analysed using a TOC/TN analyser (Shimadzu, Japan). Estimates of MBC and MBN were derived by calculating the difference between the results of the corresponding fumigated and non-fumigated analysis, divided by the extraction efficiency factor. Factors of 0.45 (Vance et al., 1987) and 0.54 (Brookes et al., 1985) were used for MBC and MBN, respectively, to account for uncompleted extraction of C and N in the microbial cell walls (Jonasson et al., 1996).

### 2.6 Microbial Activity

Soil enzyme/microbial activities were measured from samples (n = 4-6 per site) collected on six sampling occasions (March, August and October, 2014, March, May and July, 2015). All soil samples were sieved through a 2 mm sieve and analysed in triplicate.





Beta-glucosidase (BG) was determined using the method described by Eivazi and Tabatabai, (1988). After placing 1 g of soil in a 50 ml flask, 0.25 ml toluene, 4 ml of modified universal buffer (pH 6.0) and 1 ml β-D-glucoside and p-nitrophenyl-α solutions (PNG) were added sequentially and mixed by swirling. Samples were then incubated at 37 $^0$C for 1 hr, following which 1 ml of 0.5 M $CaCl_2$ and 4 ml of 0.1 M of tris (hydroxymethyl) aminomethane (pH 12) were added to halt further reactions. The supernatants were filtered and the absorbance of the filtrate measured at 410 nm using a spectrophotometer (Beckman Coulter, DU 530, UV/vis spectrophotometer). For control samples, the same procedure was followed except that PNG was added just before filtering the soil suspension instead of adding it at the beginning.

Protease activity (PRO) was determined as described by Kandeler et al., (1999). After 5 ml of sodium caseinate solution was added to 1 g soil, the samples were incubated at 50 $^0$C for 2 hours and then filtered after adding 5 ml of trichloroacetic acid solution. Alkali and Folin-Ciocalteu's reagents were added to the filtrates before protease activity was determined colorimetrically at 700 nm.

To determine nitrate reductase activity (NR), 4 ml of 2, 4-Dinitrophenol solution and 1 ml of $KNO_3$ were added to 5 g samples of soil. After incubation at 25 $^0$C for 24 hours, 10 ml 4 M KCl was added and filtered. To 5 ml of the filtrate, $NH_4Cl$ buffer (pH 8.5) sulphanilamide reagent was added, and the activity of the enzyme nitrate reductase was measured colorimetrically at 520 nm.

Total microbial activity was assayed using fluorescein diacetate (FDA), based on the method described by Schnürer and Rosswall, (1982), and later modified by Green et al. (2006). Sodium phosphate buffer (pH 7.6) and FDA lipase substrate were added to flasks containing 1 g samples of soil and incubated for 3 hours at 37 $^0$C. The fluorescein content in the filtered sample was measured at 490 nm.

### 2.7 Statistical analysis

All statistical analyses were performed using Minitab 16. All the values reported are means of three to six replicates and standard errors were included when required. To investigate the effects of flooding, we tested for significant differences between the two sites (i.e. LFS and SFS) with different hydroperiods, depending on the state of the sites in terms of water-logging (i.e. during and after inundation) over the study period. This was carried out by applying analysis of variance (ANOVA) for each flux, soil enzymatic activity, TC, TN, mineral N and microbial biomass. Functional relationships between potential environmental drivers and the fluxes of $CO_2$ and $N_2O$ were performed using linear or exponential regression models. Multiple regression analysis was used to determine the relative contribution of more than one independent environmental driver on $CO_2$ and $N_2O$ fluxes. Normality and homogeneity of the variance of all the models were checked visually from residual versus fitted plots and, when necessary, either square-root or log transformations applied. Differences were considered statistically significant when $P<0.05$, unless otherwise mentioned.





## 3    Results

### 3.1    Soil Characteristics

The relative proportion of clay is higher at the LFS, but both sites have sandy loam soil textures. Soil pH was higher at the SFS than the LFS, and increased with depth at both sites (Table 1). Porosity at the LFS was greater than at the SFS. The soil C and N concentrations were significantly higher ($P<0.001$) at the LFS site, with the greatest difference in the upper soil layers. At both sites, C and N decreased with soil depth, but more gradually at the SFS (Fig. 2a, b).

### 3.2 Rainfall, water depth, redox potential and air temperature

Depending on the timing and duration of flooding the following periods could be identified, as indicated on Fig. 3. Period A: LFS and SFS flooded; Period B: LFS flooded; Period C: neither flooded; Period D: LFS flooded; Period E: neither flooded. Thus, there was no equivalent during 2014/2015 for the Period A (the SFS was not flooded in 2015).

Air temperature followed the typical seasonal pattern (Fig. 3a) for this latitude with the highest values during the summer months (June-August) and the lowest during the winter months (December-February). The values ranged from a high of 23 $^0$C to a low of -3.9 $^0$C with an average of 7.86 $^0$C over the 2-year study period. Rainfall was very variable over the study period (Fig. 3b) with some of the highest rainfall amounts occurring during the warmer months. However, the numbers of rainfall days were higher in the cooler period. The mean annual rainfall (866 mm) at our site during the study period was considerably above the 30 year value (756 mm) measured at the Dublin Airport station.

Standing water was only present at the SFS site for ~one month and only in the winter of 2014 and reached a depth of 5 cm (Fig. 3c). In contrast, standing water was present for ~six months at the LFS site in 2015, and reached a depth of ~25 cm in 2014. Whilst standing water at the LFS site was largely associated with the winter periods, it also extended into the spring and autumn periods in 2014/2015 (Fig. 3c).

Variations in soil redox potential were broadly correlated with the periods of standing water, with the lowest values occurring in the LFS site in April, 2014 and March 2015 (Fig. 3d) and reflect the reducing conditions associated with longer periods of inundation. In contrast oxidising conditions were always found at the SFS site and these were consistently higher (more positive) than those found at the LFS site (Fig. 3d). Measurements made on the water column, however, indicated that this was always oxidising and had a low but positive redox potential (Fig. 3d).

### 3.3 Seasonal variations in $CO_2$ and $N_2O$ fluxes

The results are presented with reference to the periods A-E, which represented the state of the sites in relation to water availability. The $CO_2$ (Fig. 4a) and $N_2O$ (Fig. 4b) emissions showed a marked seasonal variation with the highest $CO_2$ values during the summer months (June-September) for both sites, which were correlated with the lowest soil moisture values (Fig. 4c) and the highest temperatures (Fig. 4d). The highest $CO_2$ emissions were



observed from the LFS site during this period (C, E) in each year (Fig. 4a). However, somewhat higher values were found in 2015, particularly for the SFS site (Fig. 4a). The lowest emissions were observed during the winter months when there were no significant differences (P=0.768) between the LFS and SFS sites, even though there was standing water at the LFS site (Fig. 3c).

Whilst there were no differences in $CO_2$ emissions between the LFS and SFS during period A (Fig. 4a), an increase in emissions in period B was observed that correlated with the period of declining water levels (Fig. 3c). Within period C, there was also evidence of decreasing $CO_2$ emissions (Fig. 4a) at the lowest (<20%) soil moisture levels (Fig. 4c). The initial part of Period C is characterised by low and similar values of $CO_2$ emissions for each site that are not appreciably different (P= 0.956) from those of the preceding period B. At the

LFS, $CO_2$ emissions during this period ranged from 16.94 to 498.18 mg $CO_2$-C $m^{-2}$ $hr^{-1}$, whereas at the SFS, the values ranged from 16.68 to 429 mg $CO_2$-C $m^{-2}$ $h^{-1}$, and were, for June-September, 2014, significantly different (P< 0.034).

During period D, the highest $CO_2$ emissions were recorded in the first month of inundation at both the LFS and SFS and declined in both sites over time. In the second growing period (Period E), $CO_2$ emissions ranged from

53.32 to 477.26 mg $CO_2$-C $m^{-2}$ $h^{-1}$ at the LFS and from 55.68 to 343.44 mg $CO_2$-C $m^{-2}$ $h^{-1}$ at the SFS. The difference between the two sites during part of this period (May-August, 2015) was significant (P= 0.013).

Estimated values of $Q_{10}$ were 2.49 and 2.08 at the LFS and SFS, respectively.

Unlike the $CO_2$ fluxes, the $N_2O$ emissions generally showed no discernible pattern during the study period (Fig. 4b) and there were no systematic changes over time. No significant differences (P> 0.05) in $N_2O$ fluxes were

observed between the LFS and SFS in any of the 5 periods. In the first wetter periods (Periods A and B combined), $N_2O$ fluxes ranged from -0.210 to 0.319 mg $N_2O$-N $m^{-2}$ $h^{-1}$ and from -0.503 to 0.364 mg $N_2O$-N $m^{-2}$ $h^{-1}$ at the LFS and SFS, respectively. In period D, the $N_2O$ fluxes from the LFS showed relatively higher variations over time than from the SFS, with values between -0.203 to 0.695 mg $N_2O$-N $m^{-2}$ $h^{-1}$ for the LFS and -0.307 to 0.206 mg $N_2O$-N $m^{-2}$ $h^{-1}$ for the SFS. Higher $N_2O$ fluxes were, however, generally observed from the

LFS than from the SFS. In the growing season (Period C), SFS showed consistently higher and more positive $N_2O$ fluxes; the maximum emission recorded (0.651 mg $N_2O$-N $m^{-2}$ $h^{-1}$) was, however, from the LFS, at the end of July. $N_2O$ fluxes less than zero suggest uptake of $N_2O$, which was more common at the SFS site.

Annual $CO_2$ emissions were 8.18 and 11.24 Mg $CO_2$-C $ha^{-1}$ $y^{-1}$ from the SFS and LFS, respectively, with values for the LFS 1.4 times higher than that from the SFS. The annual $N_2O$ emissions were 1.3 times higher from the

LFS 7.09 kg $N_2O$-N $ha^{-1}$ $y^{-1}$ compared to the SFS 5.47 kg $N_2O$-N $ha^{-1}$ $y^{-1}$. In the year SFS was not flooded (May, 2014-April, 2015), the annual $CO_2$ and $N_2O$ emissions were 8.76 (SFS) and 11.5 (LFS) Mg $CO_2$-C $ha^{-1}$ $y^{-1}$, and 3.62 (SFS) and 5.49 (LFS) kg $N_2O$-N $ha^{-1}$ $y^{-1}$, respectively.

### 3.4 Relationship between $CO_2$ and $N_2O$ fluxes and environmental parameters

Independent testing of the various environmental variables showed that soil temperature, soil water content,

redox potential, and, for the LFS, water depth was significantly correlated with soil $CO_2$ fluxes. Significant



positive correlations were also found between $CO_2$ emissions and soil temperature for both SFS ($R^2 = 0.44$, P<0.001) and LFS ($R^2 = 0.56$, P<0.001), with a somewhat greater response at LFS (Fig. 5a). A significant negative linear relationship between soil moisture and $CO_2$ emission was found at the LFS ($R^2 = 0.52$, P<0.001) and the SFS ($R^2 = 0.54$, P<0.001) (Fig. 5b). Correlations between $CO_2$ emissions and redox potential were low

with $R^2$ values of 0.25 (LFS) and 0.16 (SFS), respectively, but significant at P<0.05. $CO_2$ emissions at the LFS were exponentially correlated with water depth ($R^2 = 0.45$, P<0.001) (Fig. 5c). Combinations of soil temperature and soil water content in multiple regression analysis only resulted in a small increase in explanatory power to ~ 58 and 66 % at the SFS and LFS, respectively, but no relative contribution of redox potential was found in this analysis.

No significant relationship was observed between soil $N_2O$ fluxes and any of, soil temperature, soil moisture, redox potential or water depth at the LFS. At the SFS, soil $N_2O$ fluxes did correlate positively with soil moisture ($R^2 = 0.13$, P<0.01) and soil temperature ($R^2 = 0.13$, P<0.01), but with a low explanatory power. The $N_2O$ flux was also not correlated with redox potential at the SFS.

### 3.5 Soil enzymatic/microbial activity

While there are some significant differences on different sampling dates for BG, FDA and PRO, overall they were of similar magnitude and showed similar variation at both sites (Fig. 6). For period B, BG activity was significantly lower (P = 0.017) at the LFS, however, in the second flooding period (the first D), BG was significantly higher (P = 0.001) at the LFS than at the SFS. In the later period, E, FDA activity at the LFS was significantly higher (P = 0.001) than at the SFS. In contrast, NR activities were consistently and significantly

lower (P < 0.001) at the SFS and independent of water status (standing water availability).

### 3.6 Microbial biomass and soil $NO_3^-$ and $NH_4^+$

Seasonal variations in microbial biomass (MB) appear to differ between the LFS and SFS (Fig. 7). Total MBC was generally higher at the LFS than at the SFS at each sampling period, but particularly significant (P < 0.01) for periods B, late D and early E (Fig. 7a). On only two sampling dates was MBN at the LFS slightly lower than

at the SFS; for the remaining dates, higher MBN values were observed at the LFS (Fig. 7b). MBC: MBN ratios were significantly higher (P < 0.01) at the LFS during periods of standing water (Fig. 7c).

The concentrations of $NH_4^+$ and $NO_3^-$ were generally higher at the LFS than at the SFS (Table 2). The highest concentrations of $NH_4^+$ were generally associated with periods of standing water (LFS; March 2014/2015) or immediately after (SFS; March 2014) the disappearance of standing water (Table 2 and Fig. 3c).

## 4    Discussion

The annual $CO_2$ emissions were 11.24 - 11.5 and 8.18 - 8.76 Mg $CO_2$-C ha$^{-1}$y$^{-1}$ from the LFS and SFS, respectively. Longer term flooding therefore increased, rather than reduced, the annual emissions by approximately 40 % and suggests that any increase in freshwater flooding in response to climate change could result in a significant increase in carbon dioxide emissions from these systems. The annual emissions of $CO_2$

found in this study are in line with those previously reported for floodplain wetlands (10.91 ± 0.54 Mg $CO_2$-C





ha$^{-1}$y$^{-1}$) (Batson et al., 2015), coastal plain wetlands (11.29 Mg $CO_2$-C ha$^{-1}$y$^{-1}$) (Morse et al., 2012) and occasionally (9.7 Mg $CO_2$-C ha$^{-1}$y$^{-1}$) or frequently flooded (13 Mg $CO_2$-C ha$^{-1}$y$^{-1}$) riparian forests (Jacinthe, 2015). Our results are in part agreement with other investigations of similar ecosystems that examined the impact of flooding on GHGs (Morse et al, 2012; Jacinthe, 2015; Kim et al., 2015; Marín-Muñiz et al., 2015).

Jacinthe (2015) reported that $CO_2$ fluxes during summer were larger from a riparian forest affected by floods in winter and spring than from a flood protected area. Similarly, Morse et al, (2012) found higher rates of $CO_2$ emission in the dry period from short and intermittently flooded restored wetland habitats compared to both permanently flooded and unflooded sites.

### 4.1 Relationships between $CO_2$ fluxes and water and nutrient availability

The overall negative relationship between $CO_2$ emissions and soil moisture (Fig. 5b) suggests that flooding or high soil water availability through the creation of low oxygen conditions impedes the decomposition processes that lead to $CO_2$ production. Standing water would also act as an additional constraint on annual emissions by acting as a physical barrier to gaseous diffusion. However, somewhat paradoxically, larger annual $CO_2$ emissions were asssociated with the site with the longer flooding period. However, the highest $CO_2$ emissions

*and* the period when the differences in $CO_2$ emissions between the two sites were greatest occurred in the summer season after the disappearance of standing water, when the soil was better oxygenated (Fig. 3d). No significant differences in $CO_2$ fluxes between the LFS and SFS were observed during other parts of the year. Whilst the presence of standing water during the autumn/winter months could constrain $CO_2$ emissions at the LFS by acting as a gaseous barrier, the similar values found for the SFS for the same period indicates that this is

unlikely to have a significant impact on the annual emissions. Reductions in mineralisation caused by low temperatures may be the more significant factor at these times of the year, consistent with the strong correlations between $CO_2$ emissions and temperature that were observed in this study. Clearly, the specific impact of flooding on annual $CO_2$ emissions could therefore depend critically on the timing of flooding events.

Higher $CO_2$ emissions were obtained at the LFS during the drier parts of the year, when there were similar

values for soil moisture/soil temperature at both sites. This may be related to higher organic matter content and nutrient status and a generally higher microbial biomass. Compared to soils supplied with no or little organic matter/nutrients, soils that have received more organic matter are likely to emit substantially larger amounts of $CO_2$ (Winton and Richardson, 2015). The availability of organic matter is considered one of the most important factors controlling the production of GHGs in wetlands (Badiou et al., 2011). This is often derived from plant

production but can be introduced by incoming flood water. In our study, both the difference in total C and N values between the LFS and SFS sites and, specifically, the rapid decline in these nutrients down through the soil profile indicate these are derived largely from external sources, rather than from *in situ*, plant-related material. Had the carbon been contributed mainly from the plant community, similar or higher carbon contents would have been expected from the SFS. However, based on the above ground dry biomass estimations made

during the summer of 2014, these were approximately 6 fold higher at the SFS (35.51 Mg ha$^{-1}$) compared to the LFS (6.02 Mg ha$^{-1}$), indicating that the nutrients originated from outside the site and were presumably associated with drainage water. Many studies have reported an increase in the release of $CO_2$ via soil respiration as a result of the increased input of organic matter, particularly in wetlands (Samaritani et al., 2011; Winton and





Richardson, 2015). The source of the organic matter will be dependent on the hydrological connectivity of the wetland to the upstream land use system in addition to plant litter derived from wetland plants (Hernandez and Mitsch, 2007). Given that the C fluxes observed in this work are determined largely by external nutrient inputs, this adds to the growing body of evidence that biogeochemical processes, including greenhouse gas emissions, in floodplain wetlands are predominantly determined by offsite/catchment-related events (Batson et al., 2015). Whilst the results of our study indictate that the availability of organic matter has the major impact on the total $CO_2$ emissions, annual (seasonal) variations are related to on-site soil water content and temperature, which presumably impact via their effect on the decomposition of organic matter (Curiel Yuste et al., 2007).

**4.2 Dependence of $CO_2$ fluxes on soil temperature**

Our results also provide some insights regarding the potential effect of temperature on $CO_2$ production. A larger proportion of the variability in $CO_2$ fluxes (Fig. 5a) was explained by temperature at the LFS than at the SFS, although the temperatures were essentially identical at these two closely adjacent sites (Fig. 4d). The greater abundance of soil carbon and nitrogen, accompanied by a generally higher microbial biomass suggests substrate availability at the LFS was greater for soil microbial processes (Wang et al., 2003). The higher $Q_{10}$ values at the LFS (2.49) compared to the SFS (2.08) also suggest differences in the microbial populations at the two sites. The higher $Q_{10}$ values for $CO_2$ emissions at the LFS indicates that longer term flooded sites could be more sensitive to climate change related warming (Zhou et al., 2014).

The subtle effects of temperature on its own are illustrated by the $CO_2$ fluxes from the LFS in two episodes near the beginning and end of the flooding period (Period B and D) when there was standing water and little variation in soil moisture status (Fig. 4a). Firstly, just before the end of the first flooding period (Period B, terminated April 2014), a decrease in standing water level coincided with an increase in temperature that resulted in a pulse of $CO_2$ emission. Secondly, a few days after the beginning of the second flooding event (Period D, initiated October, 2014), a $CO_2$ efflux was observed in conjunction with a temperature increase. In neither case was there a similar response in the SFS. This indicates that temperature can, at some times, have a greater impact than water availability in determining $CO_2$ fluxes. Whilst the reason(s) for these temperature-associated emissions is not clear, it could be due to release of accumulated air bubbles in the sediment (ebullition) at the LFS (Venkiteswaran et al., 2013). However, due to the higher solubility of $CO_2$, the contribution of ebullition as a mechanism for $CO_2$ emissions is often thought to be very low (Abril et al., 2005). In general, most of the higher $CO_2$ fluxes (>170 mg $CO_2$-C $m^{-2}$ $h^{-2}$) occurred when the soil temperature was above 15 $^0$C with a soil moisture of 20-40 %. Kim et al, (2015) showed an increase in $CO_2$ production with increasing temperature from incubated flooded and non-flooded boreal soil. This again highlights the importance of the timing of flooding events and also the need to consider temperature variations when investigating the impact of flooding on GHG emissions.

**4.3 Effect of redox potential on the soil $CO_2$ and $N_2O$ fluxes**

Episodic flooding and draining is the likely cause of the greater seasonal variation in the oxidation-reduction level at the LFS compared to the more limited variation at the SFS where there was only a short hydroperiod. The weak correlation between redox potential and $CO_2$, and the absence of a relationship with $N_2O$ emissions




(Section 3.4) for the two sites, suggests soil redox potential had minimal influence on the emissions of these gases. This appears to contrast with the finding of Marín-Muñiz et al., (2015), who identified redox potential and water level as the main factors controlling $CO_2$, $N_2O$ and $CH_4$ emissions in coastal freshwater wetlands.

The redox potential of the SFS and LFS sites were, however, different and the ranges at which the highest $CO_2$ emissions occurred were 220-362 mv and 145-259 mv at the SFS and LFS, respectively. The data could imply that the larger $CO_2$ fluxes were enhanced by more oxidized soil conditions. However, they were not dependent on the redox level of the soil as the highest $CO_2$ emissions at the LFS occurred in the lower end of the redox range observed. The lower redox potential at the LFS after the disappearance of standing water could be due to the free-draining nature of this site and/or the presence of a higher organic matter content. Gardiner and James, (2012) showed a marked decrease in the redox potential (more negative) as a result of organic matter addition in their wet soil microcosm study. The redox status of soil is also controlled by the availability of electron acceptors and microbial activity (Oktyabrskii and Smirnova, 2012; Hunting and Kampfraath, 2013).

No particular redox range associated with greater $N_2O$ emissions was confidently identified. The fluxes were somewhat higher when the redox value was above 249 mv at the SFS and between -232 and 228 mv at the LFS. Many previous studies have reported higher $N_2O$ emissions in flood-affected wetlands over a wide range of redox potentials (-100 to 430 mv) (Yu et al., 2001; Wlodarczyk et al., 2003; Morse et al., 2012). Marín-Muñiz et al., (2015) found an optimum range of 100-360 mv for reduction of nitrate to $N_2O$ from a coastal wetland, whereas Morse et al., (2012) reported values of 89 and 5.3 mv from rarely and intermittently flooded areas, repectively, as the conditions conducive for $N_2O$ production.

### 4.4 Relationship between water depth and $CO_2$ and $N_2O$ fluxes

Several studies have shown a negative relationship between GHGs and water level in riparian wetlands (e.g. Mander et al., 2015; Marín-Muñiz et al., 2015). Increases in water depth at the LFS during the flooding period were accompanied by a decrease in the rate of $CO_2$ emissions, perhaps due to a decrease in near-surface oxygen supply as a result of high standing water levels. Water depth has been shown to be more significant than temperature in determining the variation in $CO_2$ fluxes during the inundation period (Dixon et al., 2014). Multiple regression analysis of $CO_2$ fluxes showed a significant paraboloid relationship ($R^2 = 0.62$, P<0.001) with water depth and soil temperature combined (Fig. 8) but most of this variation is explained by changes in water depth alone ($R^2 = 0.45$, P<0.001) (Fig. 5c). Peak $CO_2$ fluxes (above 75 mg $CO_2$-C m$^{-2}$h$^{-2}$) were recorded during Periods B and D when less than 9 cm water depths were coupled with soil temperatures above 11 $^0$C. No major variations in $CO_2$ fluxes were observed when the water depths were greater than 12 cm above the soil surface. Even though some studies have shown a significant relationship between $N_2O$ fluxes and water depth (Mander et al., 2015; Marín-Muñiz et al., 2015), no correlation was found in this study (Section 3.4). Audet et al., (2013) also found no significant impact of water depth on $N_2O$ emission from a temperate riparian wetland.

### 4.5 $N_2O$ fluxes and their controlling factors

No relationship was found between $N_2O$ fluxes and soil moisture content or temperature at the LFS and only a weak relationship at the SFS, suggesting the influence of these variables on nitrification and denitrification



processes leading to $N_2O$ production from these sites is limited. Several other environmental factors such as pH, oxygen concentration and nitrogen availability are known to affect the production of $N_2O$ (Ullah and Zinati, 2006; Van den Heuvel et al., 2011; Burgin and Groffman, 2012). In this study, the lower pH, as well as a higher nitrogen availability and higher MBC at the LFS would be expected to favour $N_2O$ emissions (Liu et al., 2010;

Van den Heuvel et al., 2011). However, apart from the constantly positive, and higher, $N_2O$ fluxes observed at the LFS for many of the sampling dates in period D, no appreciable differences in $N_2O$ emission were detected between the two sites. The generally higher $N_2O$ emissions from the LFS in period D might be associated with the persistent anaerobic conditions that would have favoured a transient reduction of $NO_3^-$ to $N_2O$. Higher soil $NH_4^+$ concentrations under largely anoxic conditions (Periods A, B and D) and its progressive decrease through

the aerobic period (Periods C and E) in the two sites (Table 2) indicate increased nitrification and ammonia oxidation that may lead to $N_2O$ production as aeration of soil and gas diffusion improves (Firestone and Davidson, 1989). This is a potential explanation for the larger $N_2O$ emissions observed at the SFS during the larger part of the first, and, in a few occasions, during the second summer period. Continuous and relatively low, or negative, $N_2O$ emissions were observed for both sites during the early part of periods C and E (i.e. after

draining of the flood water at the LFS). This may be due to increased N uptake by growing vegetation at the two sites, offsetting any tendency for water drainage to have increased soil aeration and thereby promote $N_2O$ formation. This potentially also explains the absence of $N_2O$ fluxes of a similar magnitude at the LFS during the summer where, unlike at the SFS, the immediate development and recovery of grasses, was likely impeded by the preceding episode of prolonged water-logging (Steffens et al., 2005).

The annual $N_2O$ emissions estimated in this work (7.09 and 5.47 kg $N_2O$-N $ha^{-1}$ $y^{-1}$ in the LFS and SFS, respectively) are within the range of emissions obtained in restored temperate wetlands in Denmark and The Netherlands (Hefting et al., 2003; Audet et al., 2013). Fisher et al., (2014) reported higher annual $N_2O$ emissions (4.32 kg $N_2O$-N $ha^{-1}$) from flood prone riparian buffer zones in Indiana, USA, compared to unflooded buffer zones (1.03 kg $N_2O$-N $ha^{-1}$).

In terms of the contribution of each gas to the global warming potential (GWP), expressed as $CO_2$-equivalents, $N_2O$ contributed only 15 to 17 %, to GWP, which is low compared to the $CO_2$ contribution of 83-85 %. We have not assessed the contribution of $CH_4$ gaseous emissions, although other findings report a high contribution of methane to the GWP from floodplains and freshwater wetlands (Altor and Mitsch, 2006; Koh et al., 2009). However, this may not always be the case and Jacinthe, (2015) showed that some terrestrial riparian ecosystems, which were exposed to different flooding frequencies, routinely acted as a strong sink for $CH_4$, except for a

small contribution in emissions from permanently flooded sites.

## 5    Conclusion

This study provides evidence that the interaction of a grassland ecosystem with the hydrologic regime, impacts on the annual emissions of greenhouse gases. Flooding duration affected the dynamics of $CO_2$ and, to a lesser

extent, $N_2O$ fluxes. Total emissions of $CO_2$ increased with flooding duration, with longer term (~6 months) flooding associated with higher annual emissions than the shorter term (~2-4 weeks) flooding. Temperature and soil water content are identified as the most important factors controlling the seasonal pattern of the $CO_2$ fluxes,





especially in the longer term flooded site. The higher emissions from the longer term flooded site is likely linked to the higher inputs of organic materials/nutrients, and associated increases in microbial biomass and possibly changes in the microbial populations. In contrast, no individual environmental parameter, or any combination of them, was found to have a major influence on the emissions of $N_2O$. However, flooding enhanced $N_2O$

production during the period in which water was standing on the longer term flooded site, likely due to enhanced denitrification, demonstrating the probable influence of inundation on the dynamics of $N_2O$ from coastal freshwater grasslands. The controlling mechanisms underpinning the observed $N_2O$ fluxes are not clear. However, the information obtained from the current study indicates that the contribution of $N_2O$ emissions to global warming would be minimal. A more extensive study of the effect of specific hydrologic patterns

(flooding frequency, timing and duration) on the dynamics of GHGs, including $CH_4$, would be required to better assess the global warming potential of flood-affected ecosystems.

**Acknowledgment**

This project is part of the Earth and Natural Sciences Doctoral Studies Programme, which is funded by the Higher Education Authority (HEA) through the Programme for Research at Third Level Institutions, Cycle 5 (PRTLI-5) and is co-funded by the European Regional Development Fund (ERDF). We wish to thank BirdWatch Ireland in Wicklow for providing the study site for this research and their support through the process of site formation.




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



Tables and Figures

Table 1. Soil properties (0-10 cm) in the LFS and SFS. Values are mean of N=4-6.

| Soil properties | LFS | SFS |
| --- | --- | --- |
| Sand (%) | 64.40 | 69.00 |
| Clay (%) | 16.00 | 6.50 |
| Bulk density (g cm$^{-3}$) | 0.73 | 1.02 |
| Porosity | 0.73 | 0.61 |
| pH | 4.79 | 5.28 |
| Total Carbon (g kg$^{-1}$) | 204.70 | 120.00 |
| Total Nitrogen (g kg$^{-1}$) | 38.50 | 10.70 |
| C to N ratio | 5.31 | 11.21 |

Table 2. Soil $NH_4^+$ and $NO_3^-$ concentrations at 0-5 cm depths (mg N kg$^{-1}$) for different sampling dates in 2014
10  and 2015.

| Date (Period) | LFS | | SFS | |
| --- | --- | --- | --- | --- |
| | $NH_4^+$ | $NO_3^-$ | $NH_4^+$ | $NO_3^-$ |
| 04 March, 2014 (B) | 30.49 | 0.99 | 25.63 | 0.52 |
| 30 April, 2014 (C) | 12.93 | 0.30 | 4.37 | 0 |
| 19 August, 2014 (C) | 12.47 | 0.28 | 16.85 | 0.29 |
| 08 March, 2015 (D) | 25.80 | 0 | 17.66 | 0 |
| 28 May, 2015 (E) | 18.24 | 2.28 | 15.88 | 0.12 |
| 17 July, 2015 (E) | 13.08 | 0.51 | 12.11 | 0.24 |



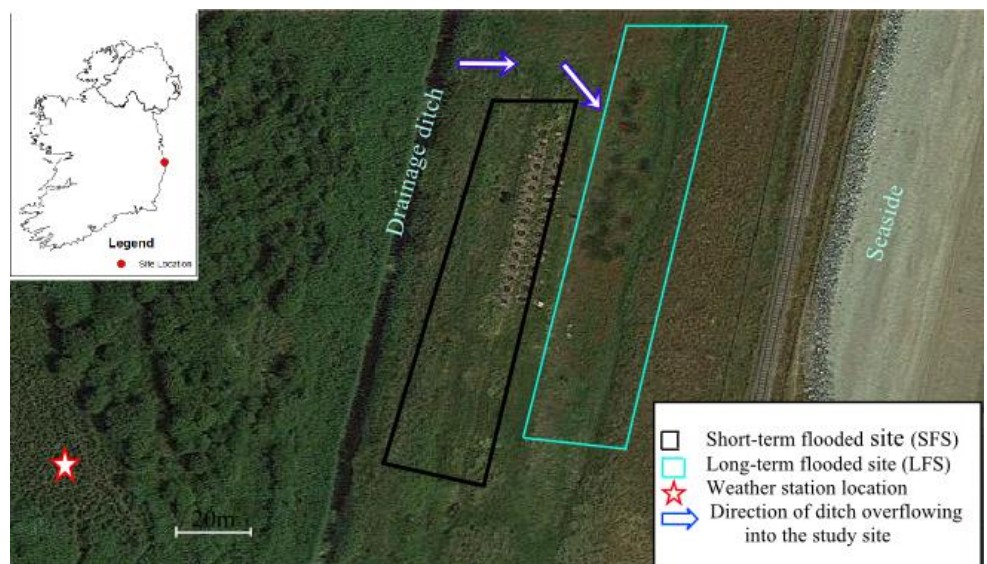

Figure 1 Aerial photograph of part of East Coast Nature Reserve at Blackditch Wood, County Wicklow, showing the study sites and the direction in which water flows during flooding. Inset shows map of Ireland with site location indicated.

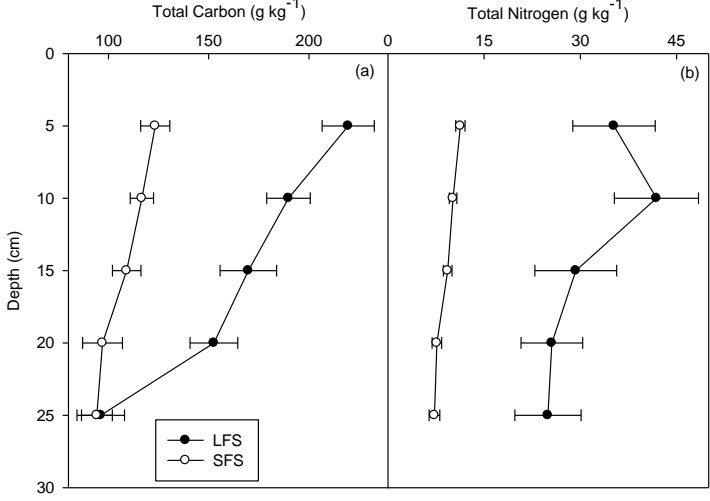

Figure 2 Profile of (a) total carbon, and (b) total nitrogen in the LFS and SFS. Mean values (n=6) are from each level within a profile and the horizontal bars represent the standard errors.





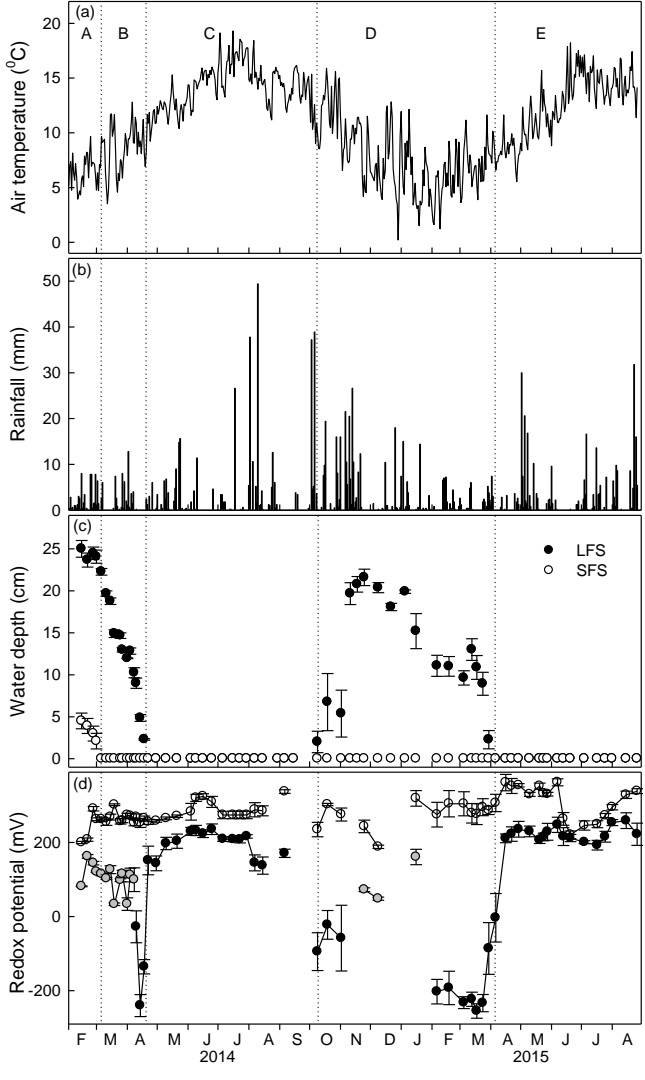

Figure 3 Variations in (a) daily air temperature, (b) daily rainfall, (c) average water depth above the soil surface, and (d) redox potential at the study sites. Open circles represent values redox from the ground surface of the SFS, black filled circles represents values from the soil surface of the LFS and grey filled circles show values

5      from the water surface while LFS was flooded to greater than 10 cm depth. Period A: LFS and SFS flooded; Period B: LFS flooded; Period C: neither flooded; Period D: LFS flooded; Period E: neither flooded. Thus, there was no equivalent of period A during late 2014 – early 2015.



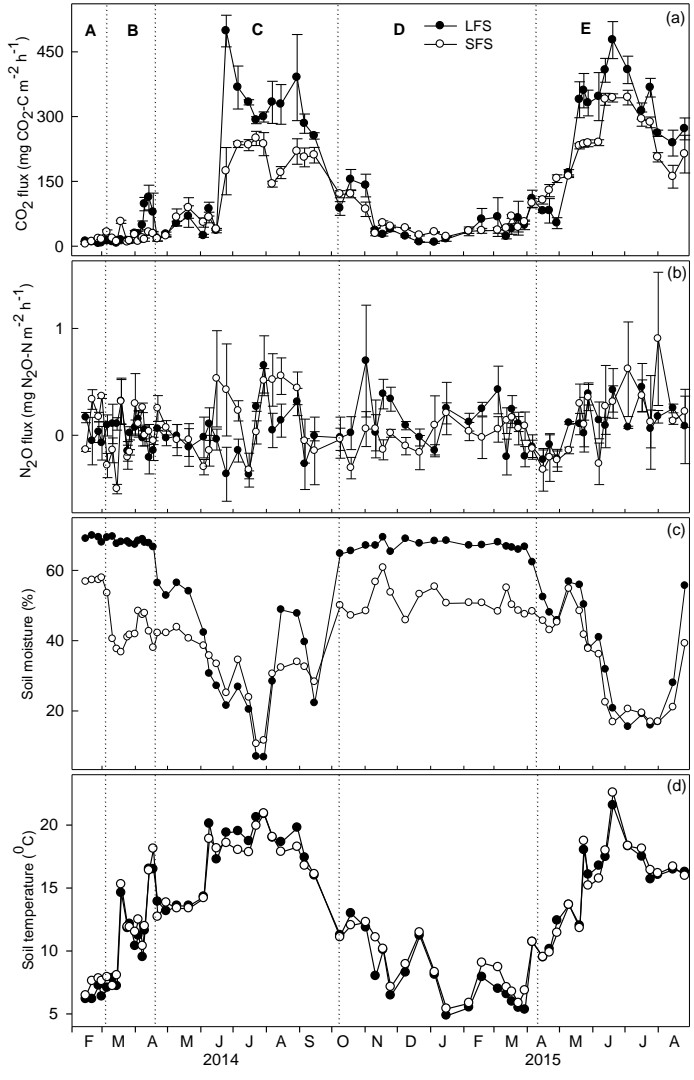

Figure 4 Variations over time in (a) $CO_2$ fluxes, (b) $N_2O$ fluxes (c) soil moisture and (d) soil temperature in each site.





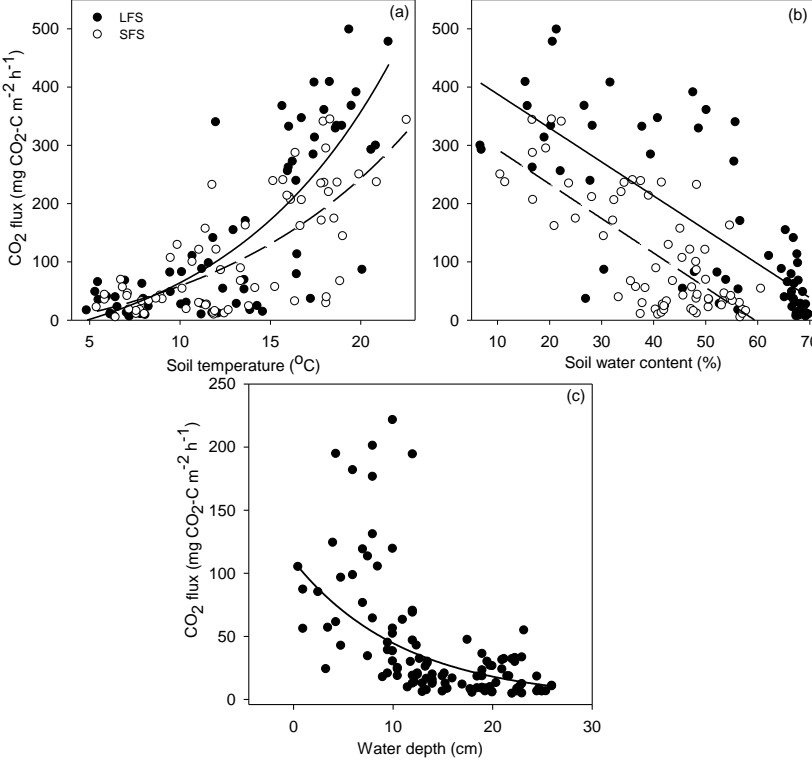

Figure 5 Relationships between $CO_2$ fluxes and (a) soil temperature, (b) soil moisture, and (c) water depth at individual collar position at the LFS. The lines at (a) represent the best fit regression y = 50.69 (exp0.09*T), $R^2$ = 0.56, P<0.001 and y = 53.62 (exp0.07*T), $R^2$ = 0.44, *P<0.001* for LFS and SFS, respectively. The lines at (b) represent the best fit regression y = 446.26 – 5.82*(SWC), $R^2$ = 0.52, *P<0.001* and y = 351.95 – 5.92*(SWC), $R^2$ = 0.54, *P<0.001* for LFS and SFS, respectively. The relationship at (c) is represented by y = 109.65 (exp (-0.04*WD)), $R^2$ = 0.45, *P<0.001*. Values in (a) and (b) are means for n=3-4.





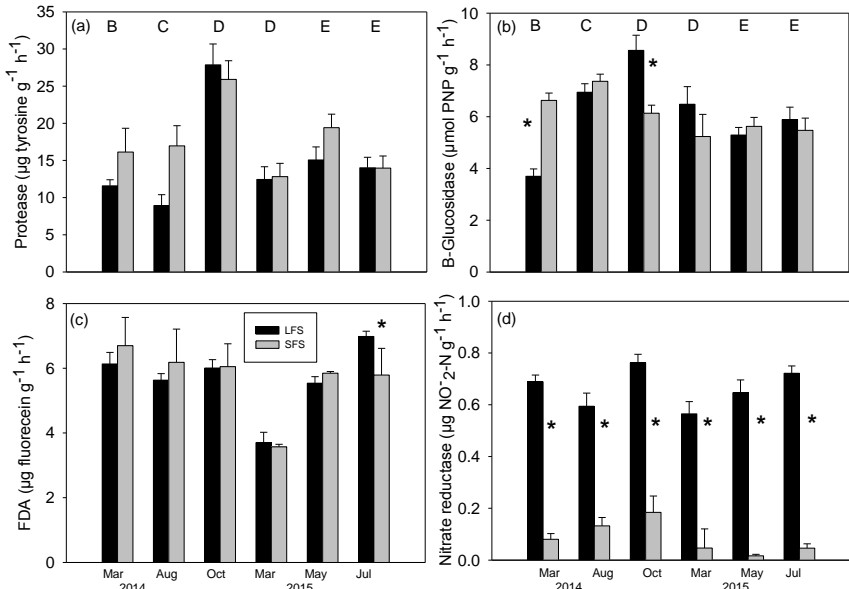

Figure 6 Soil enzymatic activities of (a) protease, (b) beta-Glucosidase, (c) fluorescein diacetate and, (d) nitrate reductase measured at different sampling dates from the LFS and SFS. The dates correspond to Periods B, C, D, D, E and E of Fig. 3. Values are mean ± SE (n = 4-6). Asterisks indicate significant difference between the LFS and SFS for the same sampling date at *P<0.05*.



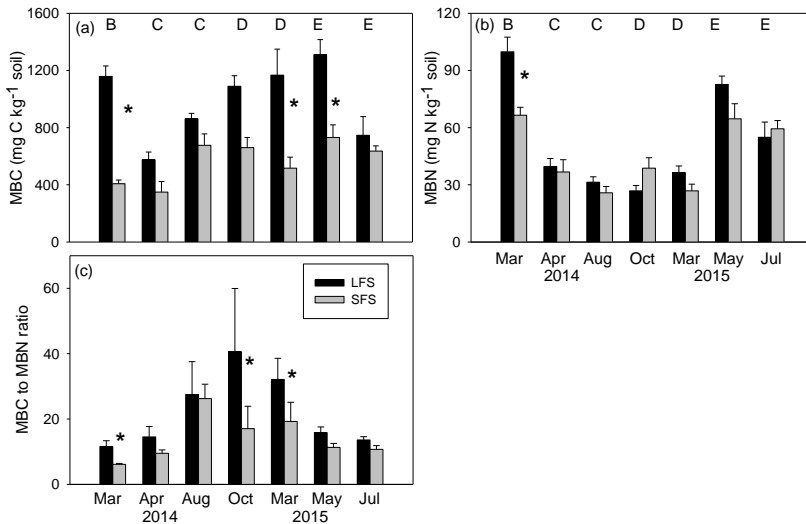

Figure 7 Seasonal variations of (a) MBC (b) MBN and (c) MBC: MBN from the LFS and SFS measured at different sampling dates. The dates correspond to Period B, C, C, D, D, E and E of Fig. 3. Error bars represent the standard errors of the mean (n=4-6). Asterisks indicate significant difference between the LFS and SFS for the same sampling date at *P<0.05*.

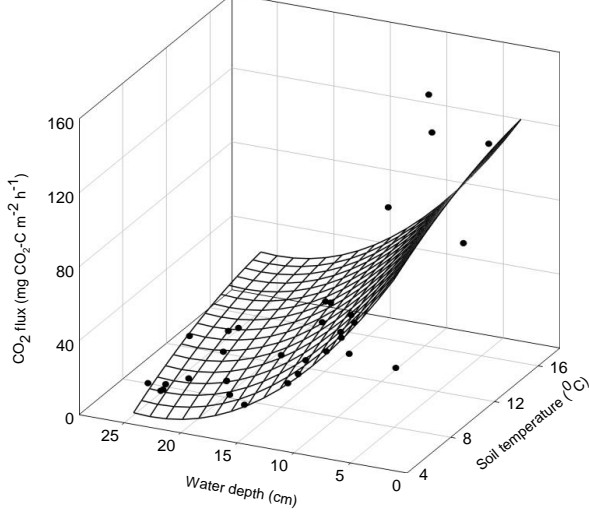

Figure 8 Dependence of $CO_2$ fluxes on water depth and soil temperature during the flooding period of LFS. A significant ($R^2 = 0.62$, *P< 0.001*) 3D Paraboloid regression is shown in the mesh plot. Values are mean of (n=3-4).