# Peer review of "Flooding-related increases in CO2 and N2O emissions from a temperate coastal grassland ecosystem"

_Biogeosciences, 2016_

## Referee Comment (RC2) · T. K. Yoon (Referee) · 13 Feb 2017

<General comments>

The study observed that the coastal grassland soil at long duration of flooding produced higher CO2 than that at short duration of flooding, probably due the exogenous substrate and nutrient loadings by flooding. Here, the somewhat contradictory results – (1) the lower CO2 under more hydric soil at the level of environmental variability in each site (e.g., Fig 5b,c); but (2) the higher CO2 under more flooding at the level of comparison between the sites in different flooding regimes (e.g. Fig 4a) – are quite interesting with the qualified data and the reasonable discussion. Personally, I have observed similar results from a nitrogen mineralization study in a temperate forested wetland (refer to dx.doi.org/10.3390/f6092941) and agree to the significance of the substrate and nutrients availability enhanced by flooding in soil biogeochemical processes as well as the general regulation of temperature and soil moisture. Therefore, the study provides a citable case regarding flooding-related GHG emissions.

However, the study design and results may not relevantly reflect the background (frequent flooding by climate change) and the implication of the study. In my understanding, the enhanced CO2 emission by flooding of the study was resulted from local hydrology and topographic factors (e.g., distance from the ditch) rather than regional climatic factors. The study design, LFS vs. SFS represented the different flooding regimes in response to topography, not climatic events. If the authors were interested in the interactions between CO2 flux and flooding in response to climate change, authors could compare interannual differences of CO2 flux. For example, at the level of interannual comparison, CO2 flux in SFS during the period C, in which a prior flooding (period A) had occurred, was lower than that during the period E without prior flooding. In SFS, 2014 was more flooded year in response to interannual climatic variability; however, CO2 emission was rather reduced in contrast to the authors' point of view. Therefore, I agree that "longer term flooding therefore increased, rather than reduced, the annual emissions by approximately 40 %" (P10 L19), in terms of the site hydrology affected by the topographical variation; whereas I reject that "any increase in freshwater flooding in response to climate change could result in a significant increase in carbon dioxide emissions from these systems" (P10 L20), according to the reduced CO2 in SFS, 2014. The flooding related increases in CO2 emission of this study could indirectly imply the relationship between frequent flooding by climate change and regional GHG budget, however, may provide little direct insights.

Moreover, as the editor and another reviewer already mentioned, lack of CH4 observations would be a critical limitation of the study which aimed to account GHG emission in especially flooding-related condition. Authors should put efforts into justifying the exclusion of CH4 with persuasive statements and/or supporting materials (references or

original data). I assume that CH4 was not investigated because the gas analyzer available to the authors does not support CH4 detection. Perhaps, CH4 emission might not be critical in the studied site where dominant CO2 emission occurred during the dry, growing season. Nevertheless, authors should clarify the study without CH4 measurements could provide complete, independent, and valuable knowledge in flooding related CO2 and/or GHG fluxes. I suggest presenting preliminary CH4 data with a minimum number that address the function of CH4 fluxes in the GHG budgets of the sites, if available.

In sum, the study presents interesting and valuable findings; however, two critical concerns, 1) somewhat irrelevant interpretation in the context of climate change and 2) lack of CH4 measurements remain. I expect the authors would improve the manuscript successfully in response to my queries.

<Specific comments>

Title and Abstract

P1 L1, P1 L22, P14 L28: Was N2O emission increased by the flooding? I could not find an evidence supporting the higher N2O emission at more flooded condition.

P1 L22, P12 L2, P14 27: There is no direct evidence supporting the changes in microbial population by flooding. The difference in Q10 values is a weak evidence.

Methods

P4 L19: In my experience, at field, a hydric soil with low bulk density can be easily compacted by investigators who stand on the soil for measurement; consequently, the compaction can physically facilitate the gas evasion from the adjacent soil, resulting in biases in measurement. Have the authors considered this issue in the measurement?

P4 L36: Were there specific QA/QC procedures for the PAS analysis, such as calibration and maintenance?

P5 L9–10: Why were the annual CO2 and N2O emissions estimated for two pseudo-different, mostly overlapped periods? I know the study only covered one and half year. Authors might attempt to provide inter-annual values. However, the two periods (Feb 2014–Feb 2015 vs. May 2014–Apr 2015) overlapped too much; therefore, the values in the two periods must be analogous. Why were the values from Apr 2015 to Aug 2015 excluded in the annual estimates?

P6 L28: Could the wet soils be sieved through a 2 mm sieve? In my case, wet soils for enzyme activity analysis were sieved through an 8 mm one. I assume a 2 mm sieve seems too fine to sieve wet soils.

Results

P8 L20: If authors were interested in the seasonal variation at each hydroperiod, the averaged values of CO2, N2O, and other environmental variables for each hydroperiod (from period A to E) could be presented in a table or a figure (bar graph) with statistical tests. In addition, annual estimates of CO2 and N2O and averaged values for all periods can be included in the table or the figure.

P9 L 21: Relationships between CO2/N2O fluxes and environmental variables (e.g., soil temperature, soil water content, water depth, redox potential, and probably microbial variables) could be presented in a table with various simple linear, multiple linear, and nonlinear regressions.

P9 L29: I am not sure whether the water depth could be a relevant independent variable for CO2/N2O modeling because the water depth data are available only for flooding period in the LFS. In other words, even in the LFS, the relationship between water depth and CO2/N2O does not cover the high CO2 emission during the growing (temperature higher than 15 °C), non-flooded (water depth lower than zero) season.

Discussion

P10 L18: The ranges of annual estimates, which were calculated from the two points

with pseudo interannual replicates, are meaningless.

P11 L17: I agree the substrate and nutrient loading by flooding could be a main driver of the enhanced CO2 emission. However, I am not sure that the vertical profile of soil C and N could be an evidence for the statement. Could you provide references or another evidence supporting the higher surface soil C driven by external sources?

P11 L34: The dependence of CO2 emissions on soil temperature is generally observed in thousands of soil CO2 studies and unquestionable today. I think the discussion can be shortened. In addition, Q10 can be an indicator of the sensitivity to climate change; however, the Q10 values from exponential regressions with the low goodness of fit might not be reliable indicators.

P13 L15: As I already mentioned, relationship of CO2 to water depth is only limited during the period when CO2 emission was not intensive. In addition, "most of this variation is explained by changes in water depth alone" is it true? Soil temperature also explained 56% of variation in CO2. The 62% of the explanatory power for temperature and soil water depth dependent model and the 45% of explanatory power for the soil water depth dependent model do not mean the small explanatory power for the temperature dependent model. I suggest that the soil water content may substitute the water depth of the model.

P14 L2: Could the growing vegetation directly uptake N2O? Please povide an evidence or reference.

Tables 1 and 2: Present statistical differences in the variables between the two sites and standard errors, too.

Figure 1: I suggest adding photo for flooded and non-flooded period of the LFS and SFS.

Figure 4: Is the soil water content volumetric or gravimetric?

<Technical comments>

P3 L1: also be contributory factors controlling -> control

---

## Author Comment (AC1)

**First we would like to thank the reviewer for the comments (in italics; responses in bold).**

*Recognition of these kinds of small scale features in landscapes, such as ephemerally flooded spots or zones is important for improving local or global greenhouse gas budgets. Role of wet spots in forest CH4 emissions is an example of equal research line. An interesting question is how to assess the spatial extent of these environments.*

**We agree with the reviewer that the spatial extent of different environments in a landscape is an important issue, hence our emphasis on the contrast between the emission histories of the LFS and SFS. The key conclusion is that small scale heterogeneity (driven by topography in this instance) can result in important differences in GHG emissions.**

*It is not quite true that "it is unclear how flooding influences annual ecosystem GHG budgets, particularly in flood-prone ecosystems that experience variable periods of inundation". There are studies on CH4, CO2, and N2O fluxes from river, lake, and pond associated flooded systems.*

**We should have phrased this differently. We intended to imply that there is no consensus, as opposed to no/few studies, as to how flooding impacts on annual GHG emissions. It is correct to say that relative to other environmental settings, studies of temperate coastal systems are limited in number; on an areal basis these are particularly important in Ireland and, more widely, in Western Europe.**

*Thus, I am surprised that CH4 was not included in the study as it is likely emitted from wet soils. In addition, the study did not include ecosystem CO2 uptake and the role of vegetation and organic matter accumulation were not dealt. For these reasons I am hesitant to support publication of this manuscript.*

**Our primary objective was to test if, and to what extent, small scale environmental heterogeneity (expressed particularly in terms of differences in the duration of flooding) could result in differences in GHG emissions. For that reason we focussed on using two variables ($CO_2$ and $N_2O$), that are often the most significant GHGs in terrestrial ecosystem, and quantified the difference between the two sites.**
**We agree with the reviewer that having demonstrated the significance of small scale heterogeneity, the impact of other potential factors, including $CH_4$, could be included in subsequent analyses.**
**We also acknowledge that $CH_4$ is often a major GHG in permanently inundated soils (e.g. wetlands, hydropower dams and lakes). In contrast ecosystems where flooding is intermittent or periodic and of a shallow depth, $CO_2$ is the dominant gas (e.g. Altor and Mitsch, 2006 Ecological Engineering; Jerman et al., 2009 Biogeosciences, Morse et al., 2012 Ecological applications; Batson et al., 2014; Jacinth, 2015 Geoderma; Winston and Richardson, 2015 Wetlands). For instance, Morse et al., (2012) showed that $CO_2$ fluxes comprised 60 to 100% of the contribution (8000-64,800 kg $CO_2$-ha$^{-1}$yr$^{-1}$) to the total GHG emissions from an intermittently and permanently flooded coastal plain; in contrast, $CH_4$ fluxes ranged from -6.87 to 197 kg $CH_4$-ha$^{-1}$yr$^{-1}$. The highest emissions of $CH_4$ were from the permanently flooded site. Broadly similar findings were reported by Batson et al., (2014), where $CH_4$ contributed 0% to the total GHG emission from**

floodplain areas with different hydroperiods. Furthermore, as the major period of flooding occurred during the cooler period of the year the lower temperatures would have restricted any flooding-related emissions of $CH_4$.

We have clarified our use of the terms "GHG annual budget" and "annual emission budget". We use 'annual emissions budget' (as we did not measure uptake by the vegetation or the soil).

In terms of the role of vegetation and organic matter accumulation, we quantified the above-ground biomass of each site for one growing season and, as reported in the discussion section, the biomass at the SFS was 5-6 times higher than that of the LFS. This suggests a larger input of organic matter from autochthonous sources in the SFS, but we found higher emissions at the LFS, suggesting that the main source of the differences in emissions was not the organic matter derived from the vegetation. The higher nutrient content of the LFS therefore implies an added, presumably external, flood-water related source, as discussed in the paper.

As an aside, the higher standing biomass at the SFS also implies greater $CO_2$ uptake relative to the LFS. Thus, the difference between the two sites is likely to be even larger based on the annual GHG budget than indicated by the emissions alone.

*Materials and Methods: description and quantification of vegetation and soils organic matter are missing. Why the activities of beta-glucosidase and protease where measured?*

There are a variety of enzymatic activities that are associated with the mineralisation/breakdown of organic matter. Two of the four we chose (details provided below) are widely used as a measure of changes in microbial activity and subsequent changes in the mineralization of organic matter (Stott et al., 2010 Soil Biology and Biochemistry; Henry, 2012; Soil Biology and Biochemistry; Vranova et al., 2013 Applied Soil Ecology).

Beta-Glucosidase enzyme catalyses the hydrolysis of B-D-glucopyranosides in the final step in the degradation of cellulose, the most abundant polysaccharide in the soil, releasing simple sugars (glucose) that are available for soil microbial populations. Protease catalyses the hydrolysis of the terminal amino acids (C and N containing) of polypeptide chains releasing nitrogen that can be utilised by soil microbes.

We will add the description for the quantification of vegetation from the sites in the method section.

The actual description for soil organic matter (total C and N) is already mentioned in section 2.4 of the paper.

*Generally, citations to previous work on effect of flooding on GHGs are lacking.*

We have provided a more extensive list of reference to previous studies focussing on the most relevant ones-intermittent flooding-in the context of our study.

---

## Author Comment (AC2)

Response #2

**We would like to thank the referee for the constructive review of the manuscript. (comments in italics; responses in bold)**

*However, the study design and results may not relevantly reflect the background (frequent flooding by climate change) and the implication of the study. In my understanding, the enhanced $CO_2$ emission by flooding of the study was resulted from local hydrology and topographic factors (e.g., distance from the ditch) rather than regional climatic factors. The study design, LFS vs. SFS represented the different flooding regimes in response to topography, not climatic events. If the authors were interested in the interactions between $CO_2$ flux and flooding in response to climate change, authors could compare interannual differences of $CO_2$ flux. For example, at the level of interannual comparison, $CO_2$ flux in SFS during the period C, in which a prior flooding (period A) had occurred, was lower than that during the period E without prior flooding. In SFS, 2014 was more flooded year in response to interannual climatic variability; however, $CO_2$ emission was rather reduced in contrast to the authors' point of view. Therefore, I agree that "longer term flooding therefore increased, rather than reduced, the annual emissions by approximately 40 %" (P10 L19), in terms of the site hydrology affected by the topographical variation; whereas I reject that "any increase in freshwater flooding in response to climate change could result in a significant increase in carbon dioxide emissions from these systems" (P10 L20), according to the reduced $CO_2$ in SFS, 2014. The flooding related increases in $CO_2$ emission of this study could indirectly imply the relationship between frequent flooding by climate change and regional GHG budget, however, may provide little direct insights.*

**The reviewer correctly states that local hydrology and topographic factors explain the principal differences in hydroperiod between the two sites. However, the frequency, duration and spatial extent of flooding are highly dependent on the seasonal rainfall pattern, which is projected to vary with climate change. .**

**Given the complex interaction between the different drivers of GHG emissions, inter-annual site-dependent differences in the effects of flooding are not unexpected, as indicated by the reviewer. In the longer term, given an increase in flooding, more of the ecosystem will behave as does the LFS and the emissions are therefore likely to increase.**

*About methane……*

**Please refer to our response to the comment by referee #1**

*Was $N_2O$ emission increased by the flooding? I could not find an evidence supporting the higher $N_2O$ emission at more flooded condition.*

**There were no statistically significant differences in $N_2O$ emissions between the LFS and SFS in any of the periods studied. However, we observed generally larger/positive $N_2O$ emissions at the LFS during the second flooding period (period D).**

*There is no direct evidence supporting the changes in microbial population by flooding. The difference in Q10 values is a weak evidence.*

**The reviewer is correct that there is no direct evidence for this. An explanation for the different $Q_{10}$ values could be because of the presence of different microbial populations and might be expected given the differing flooding regimes.**

*In my experience, at field, a hydric soil with low bulk density can be easily compacted by investigators who stand on the soil for measurement; consequently, the compaction can physically facilitate the gas evasion from the adjacent soil, resulting in biases in measurement. Have the authors considered this issue in the measurement?*

**It is true that gaseous emissions could be impacted through soil compaction and we minimized any disturbance to our sampling sites during the measurements.**

*Were there specific QA/QC procedures for the PAS analysis, such as calibration and maintenance?*

**The PAS was subjected to maintenance checks and calibration following the manufacturer's guidelines before it was used for measurements. The fine-filter paper was changed every 6 months and its holder at the back of the monitor cleaned using acetone and Q-tips as per the manual instructions. The filter pad in the ventilation unit was also washed regularly with water and soap. The calibration involved zero point calibration using pure nitrogen zero air, humidity-interference calibration using water-vapour, span calibrations using known concentrations of $CO_2$ (500 ppm) and $N_2O$ (10 ppm), and cross-interference calibration.**

*Why were the annual $CO_2$ and $N_2O$ emissions estimated for two pseudo different, mostly overlapped periods? I know the study only covered one and half year. Authors might attempt to provide inter-annual values. However, the two periods (Feb 2014–Feb 2015 vs. May 2014–Apr 2015) overlapped too much; therefore, the values in the two periods must be analogous. Why were the values from Apr 2015 to Aug 2015 excluded in the annual estimates?*

**The overlapping of the two periods is a consequence of our focus on the impact of inundation on the annual emissions on the inundation periods when we estimated the annual emissions. Including the period Apr.-Aug. 2015, as suggested by the reviewer, does not significantly change the annual emission we reported. If we include this period, the annual emissions are only reduced by 6% (34% rather than 40% increase).**

*P11 L17: I agree the substrate and nutrient loading by flooding could be a main driver of the enhanced $CO_2$ emission. However, I am not sure that the vertical profile of soil C and N could be an evidence for the statement. Could you provide references or another evidence supporting the higher surface soil C driven by external sources?*

**Evidence for higher surface soil C that is driven by external sources is given in: - Bai et al., 2005, Geoderma 124, 181-192; Bailey et al., 2007, Wetlands 27, p936-950; Winton and Richardson, 2015, Wetlands 35, 969-979.**

*Could the wet soils be sieved through a 2 mm sieve? In my case, wet soils for enzyme activity analysis were sieved through an 8 mm one. I assume a 2 mm sieve seems too fine to sieve wet soils.*

**We were able to sieve wet soil samples through a 2mm sieve after temporarily leaving the cores to drain. We also manually disintegrated soil aggregates to facilitate sieving. The reason why we were able to do this could be because of the high sand content of the coastal soil at this site.**

*I am not sure whether the water depth could be a relevant independent variable for $CO_2/N_2O$ modelling because the water depth data are available only for flooding period in the LFS. In other words, even in the LFS, the relationship between water depth and $CO_2/N_2O$ does not cover the high $CO_2$ emission during the growing (temperature higher than 15 $^0C$), non-flooded (water depth lower than zero) season.*

*As I already mentioned, relationship of $CO_2$ to water depth is only limited during the period when $CO_2$ emission was not intensive. In addition, "most of this variation is explained by changes in water depth alone" is it true? Soil temperature also explained 56% of variation in $CO_2$. The 62% of the explanatory power for temperature and soil water depth dependent model and the 45% of explanatory power for the soil water depth dependent model do not mean the small explanatory power for the temperature dependent model. I suggest that the soil water content may substitute the water depth of the model.*

**We concur that water depth cannot be an independent variable for modelling the annual fluxes as flooding was restricted to a limited period of the year at the LFS. To clarify, the relationship between water depth and $CO_2$ emissions was only examined during periods of standing water (periods A, B and D). For these periods water depth was the major factor underpinning the emissions, as indicated by Fig. 8. Soil moisture during these periods was largely invariant. As discussed in the paper soil moisture ($R^2 = 0.52$) and especially temperature ($R^2 = 0.56$) were the major drivers for $CO_2$ emissions over the whole study period.**

*The dependence of $CO_2$ emissions on soil temperature is generally observed in thousands of soil $CO_2$ studies and unquestionable today. I think the discussion can be shortened. In addition, Q10 can be an indicator of the sensitivity to climate change; however, the Q10 values from exponential regressions with the low goodness of fit might not be reliable indicators.*

**We can shorten the section about temperature if this is considered to be too detailed. Although there was a low goodness of fit for estimating the $Q_{10}$ values, which is perhaps not surprising given that they are based on field measurements, the differences were significant and support the suggestion that the longer-term flooded site could be more responsive to future increases in temperature.**

*Could the growing vegetation directly uptake $N_2O$? Please provide an evidence or reference.*

**As far as we know, there is no convincing evidence that plants take up and assimilate $N_2O$, instead relying mainly on mineral forms of N (ammonium or nitrate). Growing vegetation consumes available inorganic nitrogen in the soil, and this could reduce the amount of inorganic nitrogen available for conversion to $N_2O$. There is a large body of evidence that shows that $N_2O$ emissions are often closely associated with nitrate availability in soils. Low $NO_3^-$ availability might also facilitate $N_2O$ uptake with $N_2O$, rather than $NO_3^-$, acting as the major electron acceptor in denitrification reactions (e.g. Wagner-Riddle et al., 1997).**

*Tables 1 and 2: Present statistical differences in the variables between the two sites and standard errors, too.*
*Figure 1: I suggest adding photo for flooded and non-flooded period of the LFS and SFS.*
*Figure 4: Is the soil water content volumetric or gravimetric?*

**Standard errors will be provided in the corrected version of the manuscript. The soil moisture values reported in the manuscript are volumetric. We can also include a photograph of the site if required.**

*Relationships between $CO_2/N_2O$ fluxes and environmental variables (e.g., soil temperature, soil water content, water depth, redox potential, and probably microbial variables) could be presented in a table with various simple linear, multiple linear, and nonlinear regressions.*

**All tests for the relationship between the measured variables and $N_2O$ emissions were not statistically significant and were not presented (as explained in the text). The relationship among the measured variables and $CO_2$ emissions are reported in the text. These could be repeated in a table if required.**